# Fabrication of a Plasmonic Heterojunction for Degradation of Oxytetracycline Hydrochloride and Removal of Cr(VI) from Water

**Jihui Cao** [1], **Meihua Zhang** [1], **Xinran Yang** [2], **Xiaojun Zeng** [2], **Yubo Yang** [3], **Yuanyi Li** [4], **Hehua Zeng** [1,*] and **Wei Zhao** [2,*]

1  Department of Chemistry and Applied Chemistry, Changji University, Changji 831100, China
2  School of Materials Engineering, Changshu Institute of Technology, Changshu 215500, China
3  Department of Materials and Chemistry, Suzhou University, Suzhou 215000, China
4  Jiangsu Collaborative Innovation Center of Regional Modern Agriculture & Environmental Protection, School of Chemistry and Chemical Engineering, Huaiyin Normal University, Huaian 223001, China
*  Correspondence: rainy_lily2007@163.com (H.Z.); 8201711050@hytc.edu.cn (W.Z.)

**Abstract:** A novel $Ag/Ag_2CO_3/BiVO_4$ plasmonic photocatalyst was successfully prepared by depositing Ag nanoparticles on the surface of $Ag_2CO_3/BiVO_4$ through the photoreduction reaction. Due to the existence of this novel heterojunction photocatalyst structure, not only can it prevent the photogenerated charge recombination, but the unique properties of Ag also have a great advantage in the absorption of light. The $Ag/Ag_2CO_3/BiVO_4$ photocatalyst showed good catalytic performance in the degradation of oxytetracycline hydrochloride (OTH) and removal of $Cr^{6+}$, and the degradation rate of OTH reached 98.0% after 150 min of illumination. The successful preparation of $Ag/Ag_2CO_3/BiVO_4$ was confirmed by a series of characterization methods, and the importance of $\bullet OH$ and $h^+$ radicals in the degradation of OTH was demonstrated. In addition, the photocatalytic mechanism of $Ag/Ag_2CO_3/BiVO_4$ photocatalyst was systematically studied in terms of degradation of OTH and reduction of $Cr^{6+}$. This study is of great importance for the development of novel plasmonic heterojunction photocatalysts and to meet future environmental requirements.

**Keywords:** photocatalysis; photocatalyst; photocatalytic performance; heterojunction; degradation

## 1. Introduction

Photocatalysis is an environmentally friendly technology and has become a promising method for the elimination of organic pollutants, heavy metals and medical residues [1–9]. As an efficient, environmentally friendly, and chemically stable photocatalytic material, titanium dioxide ($TiO_2$) has inestimable application potential in photocatalysis [10–15]. However, due to its wide band gap and narrow UV response range, its application is limited. Therefore, it is urgent to find a photocatalyst driven by visible light with an appropriate light response range [16–25].

Bismuth vanadate ($BiVO_4$) has a fine effect on the elimination of organic pollutants due to its unique visible light response, excellent chemical stability, and photocatalytic activity [26–29]. However, the practical application of $BiVO_4$ is greatly limited due to its poor separation ability of photogenerated carriers and weak light utilization force. Therefore, further research is needed to remedy these deficiencies [30–35].

For improving the catalytic performance of $BiVO_4$ samples, researchers synthesized heterojunction photocatalysts containing $BiVO_4$, such as $Fe_2O_3/BiVO_4$, $BiVO_4/TiO_2$, $Ag@AgBr/BiVO_4$, etc. [36–38]. For instance, Li group prepared $g-C_3N_4/BiVO_4$ photocatalyst, which showed excellent performance in methylene blue degradation [39]. Wang et al. constructed $BiVO_4/Co_3O_4$ heterojunction photocatalyst, which showed high photocatalytic performance [40]. Gao reported $BiVO_4/Bi_2S_3$ photocatalyst synthesized by anion exchange



method, which showed superior photocatalytic activity in the reduction of $Cr^{6+}$ [41]. Based on the above results, it can be concluded that the heterojunction photocatalyst has better photocatalytic activity.

Moreover, the photocatalytic performance of the photocatalyst may be improved when it is modified with precious metals [42,43]. Due to unique properties of noble metals, namely, the local surface plasmon resonance (LSPR). This feature greatly improves the utilization of light in the visible region and the recombination of charge carriers is prevented to optimize the performance of the photocatalyst. Zhao reported that $Ag/AgVO_3/RGO$ photocatalyst's performance in BPA degradation is unmatched [44]. Based on the research and learning of predecessors, we were inspired to use Ag nanoparticles to modify $BiVO_4$-based heterojunction photocatalyst to generate LSPR. In order to verify this hypothesis, flower spherical $BiVO_4$ was synthesized by one-step hydrothermal method. Then $Ag_2CO_3$ was deposited on the surface of $BiVO_4$ to synthesize $Ag_2CO_3/BiVO_4$ photocatalyst. The $Ag/Ag_2CO_3/BiVO_4$ plasma photocatalyst was further synthesized by photoreduction reaction.

The photocatalytic properties of $Ag/Ag_2CO_3/BiVO_4$ were studied by photocatalytic removal of $Cr^{6+}$ and photodegradation of OTH. The photocatalytic mechanism of $Ag/Ag_2CO_3/BiVO_4$ photocatalyst in the degradation of OTH was systematically studied. Figure 1 shows the synthesis route of $Ag/Ag_2CO_3/BiVO_4$ photocatalyst. The plasma heterojunction photocatalyst was synthesized by photoreduction reaction, which provides ideas for the development of new semiconductor heterojunction photocatalysts.

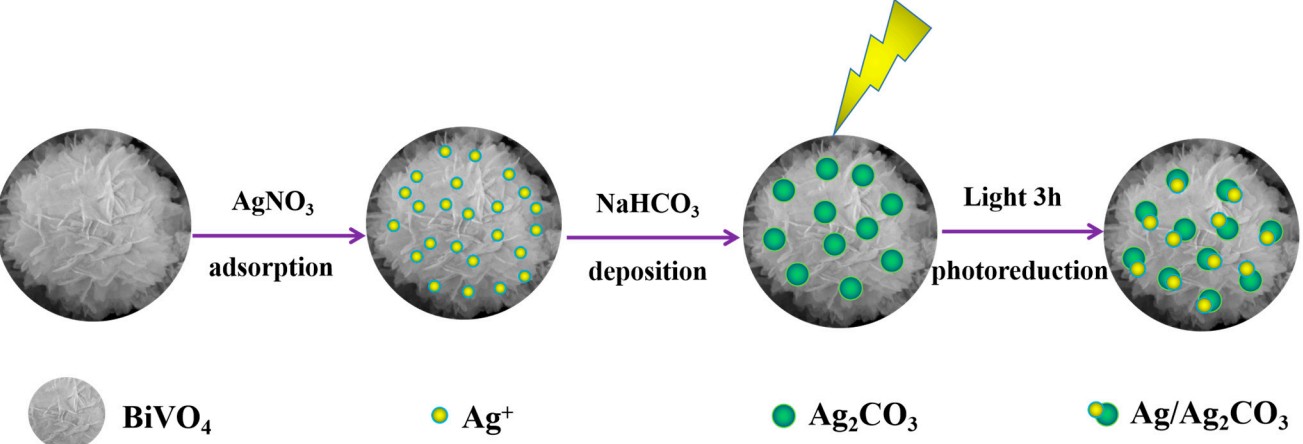

**Figure 1.** Schematic illustration of the synthesis process of $Ag/Ag_2CO_3/BiVO_4$ photocatalyst.

## 2. Results and Discussion

### 2.1. Morphology of Sample

The structure, morphology, and phase purities of $BiVO_4$, $Ag/Ag_2CO_3/BiVO_4$ and $Ag_2CO_3/BiVO_4$ samples were detected by XRD (Figure 2a). In the XRD pattern of the standard phase of $BiVO_4$ (JCPDS: 14-0688), almost all diffraction peaks are consistent with it. In addition, no Ag diffraction peak was found in the $Ag_2CO_3/BiVO_4$ sample, possibly because $Ag_2CO_3$ is highly dispersed on the $BiVO_4$ surface, or because the $Ag_2CO_3$ content is low. However, for $Ag/Ag_2CO_3/BiVO_4$, the corresponding diffraction peak is not shown in the figure because of the poor crystallinity or too little content of Ag. To further verify the presence of Ag and $Ag_2CO_3$, XPS analysis was performed on the prepared samples.

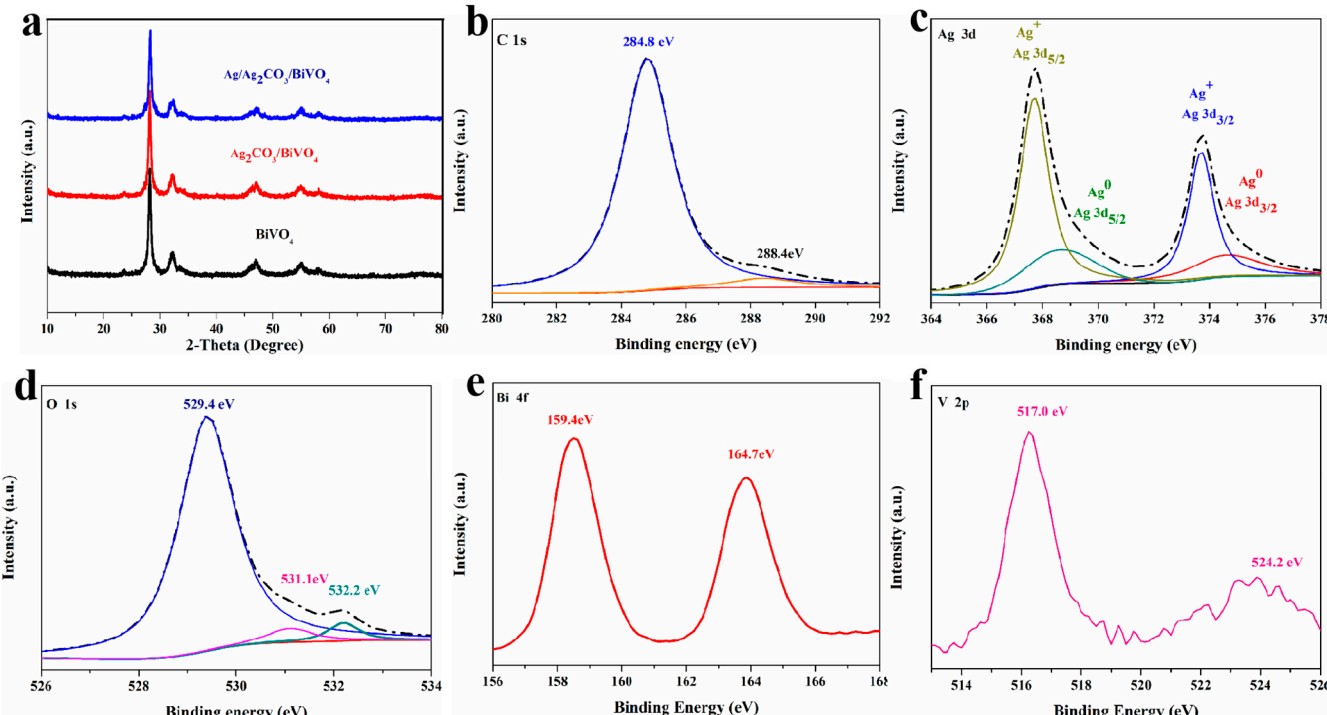

**Figure 2.** (**a**) XRD patterns of the as-prepared samples, (**b–f**) XPS spectra of each element.

In order to prove the successful synthesis of $Ag/Ag_2CO_3/BiVO_4$ and the existence of $Ag_2CO_3$, the surface chemical composition and electronic state of the samples were detected by X-ray photoelectron spectroscopy (XPS). In Figure 2b, the peak at 284.8 eV may be derived from hydrocarbon contaminants [45]. However, the main peak at 288.4 eV originates from $Ag_2CO_3$, indicating the existence of $Ag_2CO_3$ [46]. In Figure 2c, Ag 3d consists of independent peaks centered at 367.7 and 373.7 eV, corresponding to Ag $3d_{5/2}$ and Ag $3d_{3/2}$, respectively. These two independent peaks can deconvolve four bands, $Ag^+$ $3d_{5/2}$ (367.5 eV) and $3d_{3/2}$ (373.5 eV), and $Ag^0$ $3d_{5/2}$ (368.6 eV) and $3d_{3/2}$ (374.7 eV), thus confirming the existence of Ag [47]. As can be seen in Figure 2d, the peak located at 529.5 eV, indicating the presence of the Bi-O band [48]. In addition, the peaks of O 1 s at 531.2 and 532.1 eV, correspond to $CO_3^{2-}$ and -OH, respectively [49,50]. In this case, oxygen atoms undergo chemical changes on the surface of the photocatalyst, and then capture a large number of photoexcited electron-hole pairs, thus promoting the separation rate of photogenerated charges and finally improving the catalytic activity of the photocatalyst. For Figure 2e,f, the peaks of Bi 4f and V 2p located at 159.4, 164.7 eV, and 517.0, 524.2 eV are basically corresponding to $Bi^{3+}$ and $V^{5+}$, respectively.

The structure morphologies of the samples were studied by FESEM. In Figure 3a, the size of flower globular $BiVO_4$ is about 2–4 μm. The surface of $Ag/Ag_2CO_3/BiVO_4$ is rougher than that of $BiVO_4$, indicating the presence of Ag ions. Therefore, the existence of Ag or $Ag_2CO_3$ is confirmed, which is consistent with the XRD results.

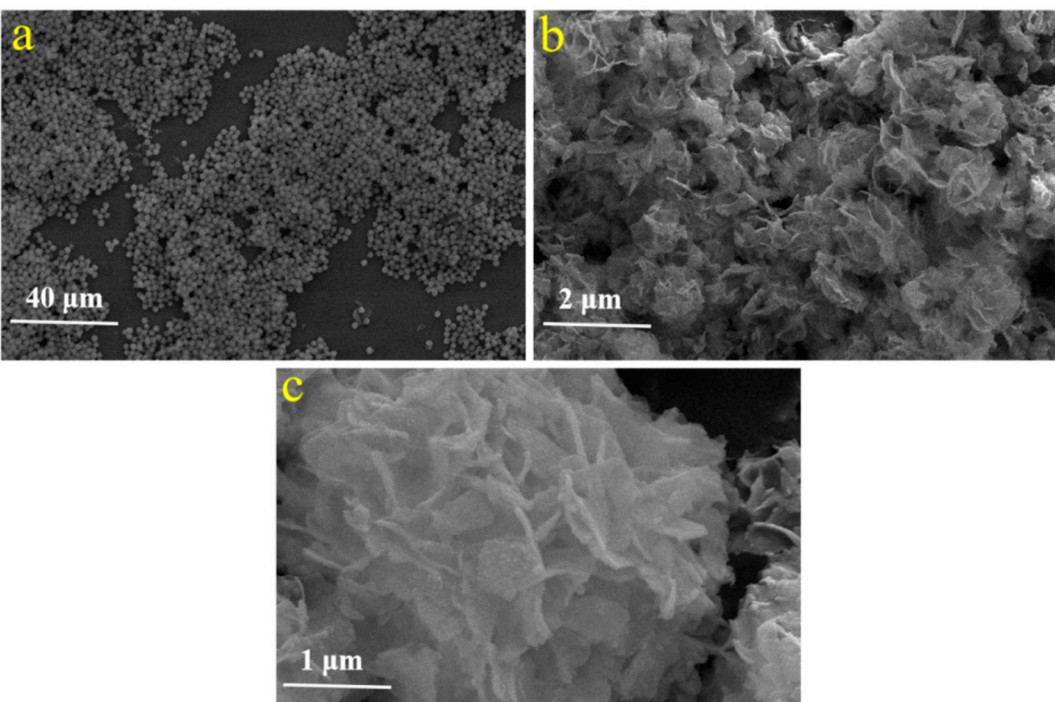

**Figure 3.** (**a**) The FESEM image of $BiVO_4$, (**b**,**c**) The SEM image of $Ag/Ag_2CO_3/BiVO_4$.

In order to understand the microstructure morphology of $Ag/Ag_2CO_3/BiVO_4$, TEM and HRTEM were performed to characterize the $Ag/Ag_2CO_3/BiVO_4$. In Figure 4a,b, the morphological characteristics of $Ag/Ag_2CO_3/BiVO_4$ can be clearly observed. From Figure 4c, the diameters of Ag and $BiVO_4$ are 0.238 and 0.312 nm, respectively, corresponding to the (111) and (121) crystal planes [51]. This unique structure will help prevent photogenerated carrier recombination.

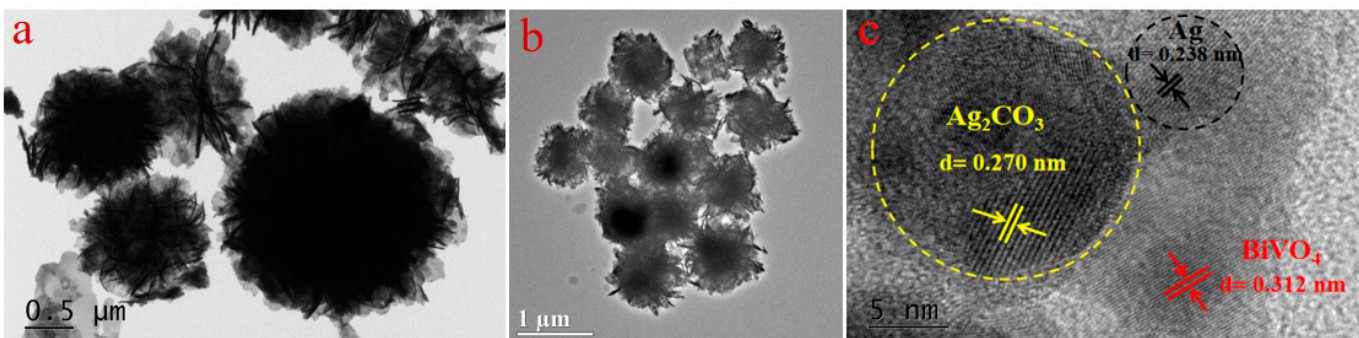

**Figure 4.** (**a**,**b**) The TEM image of $Ag/Ag_2CO_3/BiVO_4$, and (**c**) HRTEM image of $Ag/Ag_2CO_3/BiVO_4$.

### 2.2. Optical Properties

The optical properties of the samples were measured by UV–vis diffuse reflectance spectra and photoluminescence spectra. The UV–vis absorption spectra of $BiVO_4$, $Ag_2CO_3/BiVO_4$ and $Ag/Ag_2CO_3/BiVO_4$ are shown in Figure 5a. $BiVO_4$ has a strong absorption at about 500 nm, which may be band gap absorption. However, the use of light by $Ag/Ag_2CO_3/BiVO_4$ is obviously stronger than that of $BiVO_4$ and $Ag_2CO_3/BiVO_4$. This shows that $Ag/Ag_2CO_3/BiVO_4$ will have excellent photocatalytic performance under visible light irradiation. There are two main reasons for this phenomenon. First, Ag nanoparticles enhance the absorbance in the process of darkening. Second, under local surface plasmon resonance (LSPR), strong absorption occurs. Moreover, the optical absorption near band edge can be calculated according to the equation $\alpha h\upsilon = A (h\upsilon - E_g)^n$,

where α, h, υ, A and Eg are the absorption coefficient, Plank's constant, frequency of the incident photon, a constant and band gap, respectively. The index n equals 2 for direct-gap semiconductor and it equals 1/2 for indirect-gap semiconductor. As to all the samples, n equals 1/2. The band gap values of $BiVO_4$, $Ag_2CO_3/BiVO_4$ and $Ag/Ag_2CO_3/BiVO_4$ are calculated by the equation, which are 2.50, 2.43 and 2.37 eV, respectively.

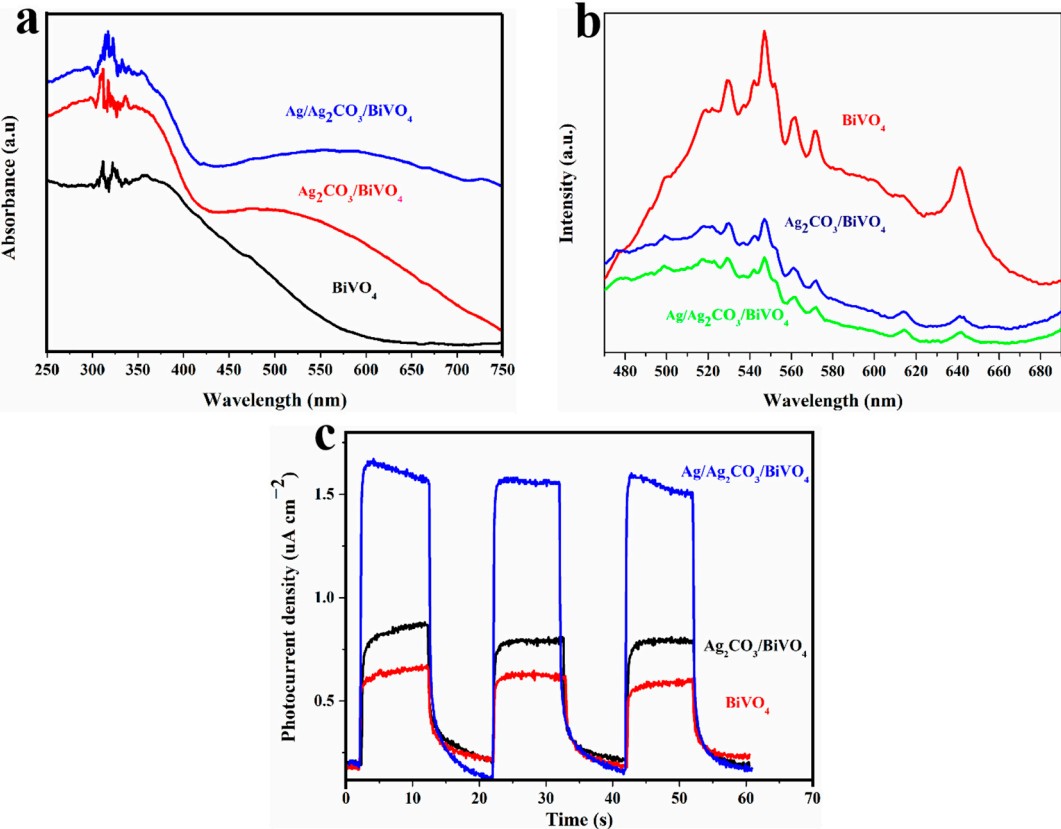

**Figure 5.** (**a**) The samples of UV—vis absorption spectra, (**b**) PL spectra, (**c**) photocurrent.

The recombination rate of photogenerated carriers is an important factor affecting the photocatalytic activity of photocatalysts. Therefore, the efficiency of photoinduced electron and hole migration was studied by PL spectrum. Simply put, a higher PL intensity means a lower number of photoelectrons and a lower hole separation efficiency. The PL spectral intensity of the prepared sample was emitted at a wavelength of 425 nm. In Figure 5b, compared with the $BiVO_4$, the PL intensity of $Ag_2CO_3/BiVO_4$ is lower, indicating that the fast carrier movement rate between $Ag_2CO_3$ and $BiVO_4$ interface is related. However, when Ag nanoparticles are deposited on the surface of $Ag_2CO_3/BiVO_4$, the PL intensity of $Ag/Ag_2CO_3/BiVO_4$ is lower, which indicates that the LSPR of Ag can enhance the absorbance and accelerate the separation rate of electron-hole pairs.

In order to study the photoresponse of photocatalyst, the separation efficiency of photogenerated carriers was analyzed by photocurrent spectroscopy. $Ag/Ag_2CO_3/BiVO_4$ has the highest photocurrent intensity, which means that the electron–hole pairs recombination rate is extremely low.

### 2.3. Photocatalytic Performance

By degrading OTH and removing $Cr^{6+}$, the photocatalytic activity of the samples was systematically studied in the range of lighting. At the beginning of the experiment, it had experienced dark adsorption for 1 h and the adsorption and desorption on the surfaces of the catalysts were balanced. Furthermore, the photodegradation experiment of OTH without photocatalyst was also carried out as blank control.

Figure 6a shows the degradation of OTH by photocatalyst. Among all the samples, only Ag/Ag$_2$CO$_3$/BiVO$_4$ showed excellent photocatalytic performance, and the degradation rate of OTH reached 98.0% after 150 min of illumination. Nevertheless, the oxidation degradation of OTH over the pure BiVO$_4$ and Ag$_2$CO$_3$/BiVO$_4$ are only 57.0% and 69.3%, respectively. According to the equation: $\ln(C_0/C_t) = k_{app}t$, $C_0$ is the initial concentration and $C_t$ is the concentration at time t, $k_{app}$ is the apparent reaction rate constant. For Figure 6b, it can be seen that the $k_{app}$ value of Ag/Ag$_2$CO$_3$/BiVO$_4$ is the highest (0.023 min$^{-1}$). Furthermore, stability is also a very important factor in photocatalytic degradation. Therefore, we studied the stability of the sample (Figure 6c). The Ag/Ag$_2$CO$_3$/BiVO$_4$ photocatalyst has the best stability and excellent performance in degrading OTH.

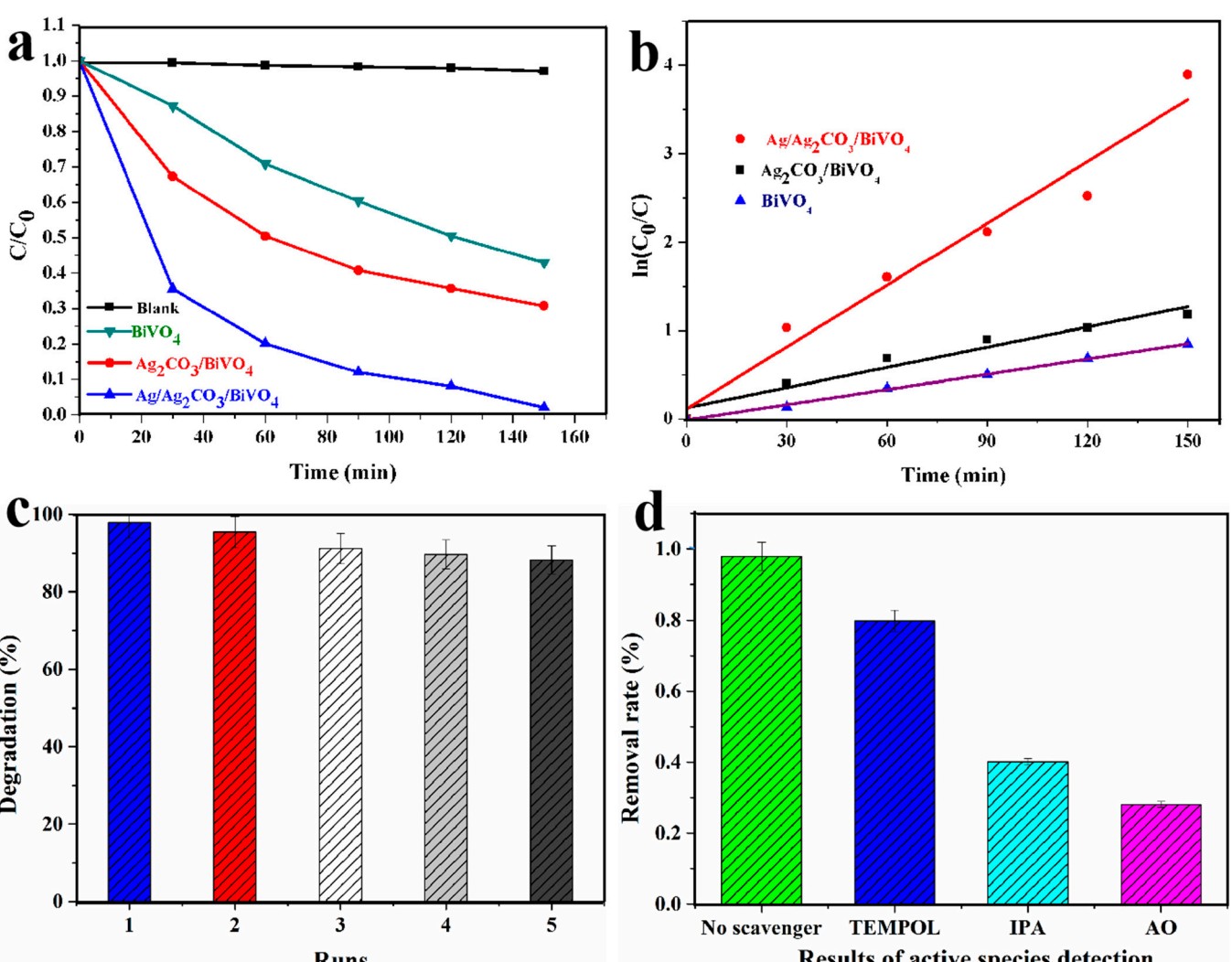

**Figure 6.** (**a**) Photocatalytic activities of the samples and (**b**) pseudo-first-order kinetics of OTH degradations, (**c**) five recycling runs of OTH degradations, (**d**) the effect of different quenchers on the degradation of OTH.

As observed in Figure 7a, the photocatalytic reduction activity of the sample was systematically investigated by photocatalytic removal of Cr$^{6+}$ under illumination. The results also show that Ag/Ag$_2$CO$_3$/BiVO$_4$ photocatalyst has the highest photocatalytic performance. In addition, we also verified the stability of the sample in the process of removing Cr$^{6+}$, as expected, showing excellent stability in Figure 7b.

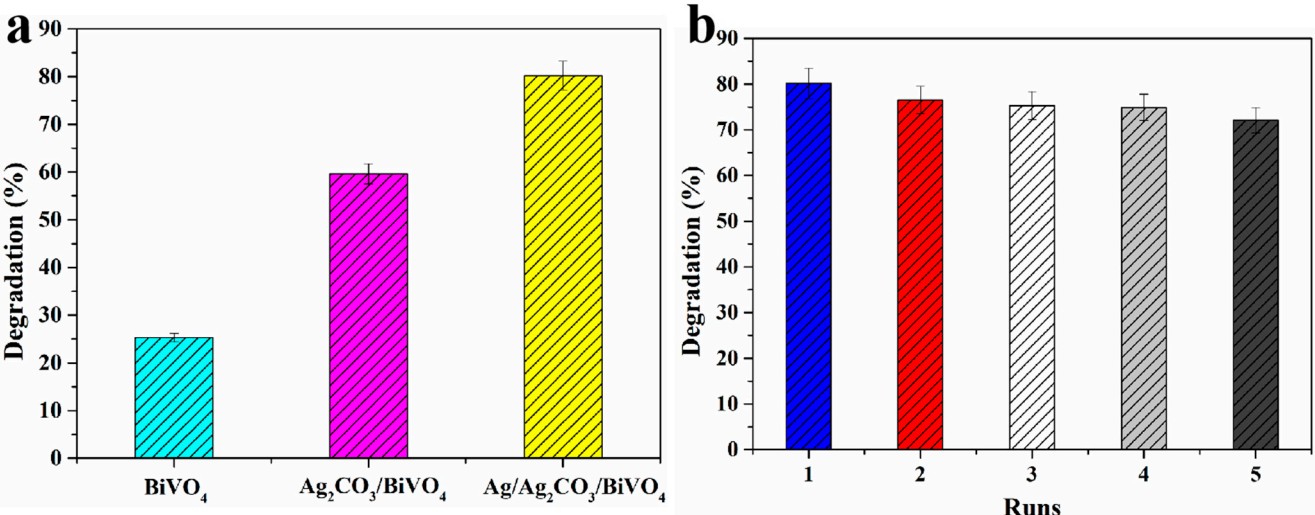

**Figure 7.** (**a**) photocatalytic reduction activities of $Cr^{6+}$ reduction, (**b**) five recycling runs of $Cr^{6+}$ reduction.

In addition, the total organic carbon (TOC) in degradation of OTH was also determined for evaluation of mineralization rate of OTH. From Figure 8, the mineralization rate of OTH in the reaction system were 34.2%, 46.8% and 68.9% corresponding to $BiVO_4$, $Ag_2CO_3/BiVO_4$ and $Ag/Ag_2CO_3/BiVO_4$, respectively, suggesting that OTH is effectively mineralized during the photocatalytic degradation process and that $Ag/Ag_2CO_3/BiVO_4$ has a higher mineralization rate.

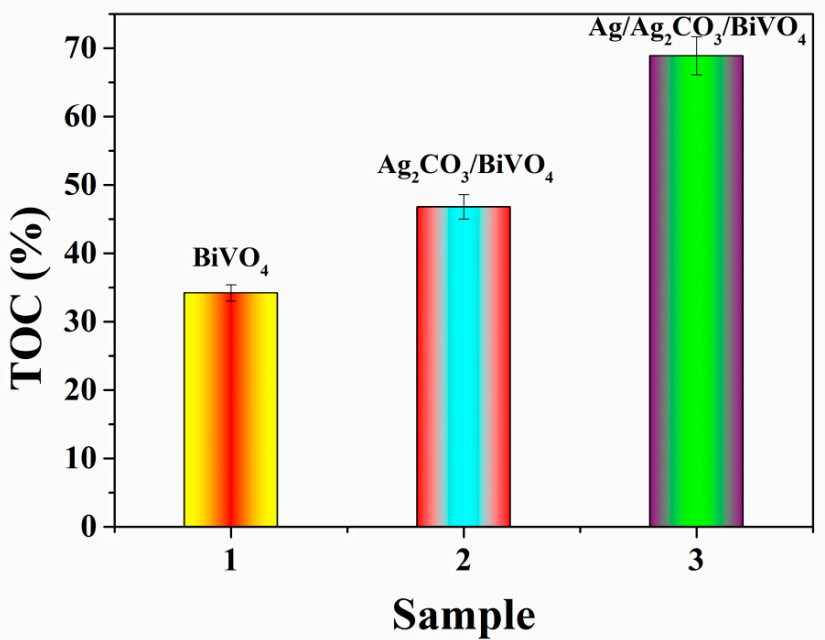

**Figure 8.** TOC removal of OTH for the different samples.

### 2.4. Photocatalytic Mechanism

Generally speaking, the role of photogenerated carriers in photocatalytic processes cannot be denied, and there are many active species in photocatalytic reactions. The trapping agents used in the experiment are ammonium oxalate (AO), isopropanol (IPA) and 4-hydroxy-2,2,6,6-tetramethylpiperidine-N-oxyl (TEMPOL), which are used to scavenge free radicals. These scavengers can capture $h^+$, •OH and $•O_2^-$, and the effects of these active species on OTH degradation were examined [52–54]. In Figure 6d, the low effect

of TEMPOL on photocatalytic degradation of OTH suggests that $\bullet O_2^-$ is not the predominant active species in photocatalytic reactions. However, when AO and IPA are added, the photocatalytic degradation rate of OTH is significantly reduced, which is inseparable from the role of $h^+$ and $\bullet OH$. Thus, $h^+$ and $\bullet OH$ are the most active species in the photocatalytic process.

As shown in Figure 9, the photocatalytic degradation mechanism of Ag/Ag$_2$CO$_3$/BiVO$_4$ photocatalyst was proposed. Under visible light irradiation, photogenerated electrons and photogenerated holes are generated on the conduction band and valence band of Ag$_2$CO$_3$ and BiVO$_4$, respectively. As shown in Equations (1) and (2):

$$BiVO_4 + h\upsilon \rightarrow BiVO_4 \ (e^- + h^+) \tag{1}$$

$$Ag_2CO_3 + h\upsilon \rightarrow Ag_2CO_3 \ (e^- + h^+) \tag{2}$$

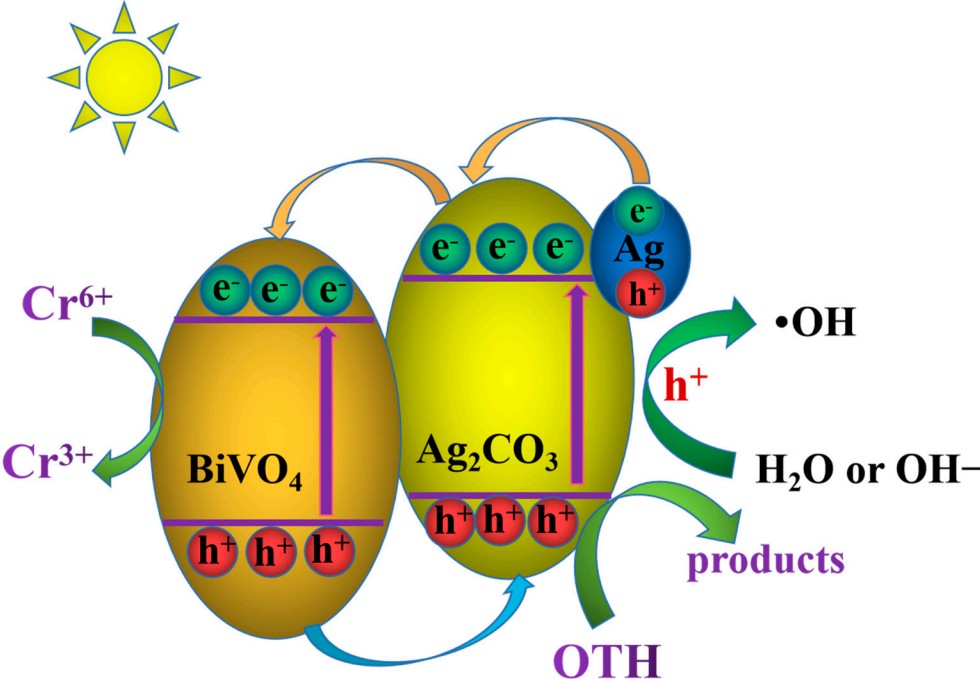

**Figure 9.** The photocatalysis mechanism of plasmonic heterojunction photocatalyst.

Since the conduction band (CB) and valence band (VB) position of BiVO$_4$ are lower than Ag$_2$CO$_3$, the conduction band of photogenerated electrons transferred from Ag$_2$CO$_3$ to BiVO$_4$, nevertheless the valence band of photogenerated holes are transferred from BiVO$_4$ to Ag$_2$CO$_3$. As shown in Equations (3) and (4):

$$Ag_2CO_3 \ (e^-) + BiVO_4 \rightarrow Ag_2CO_3 + BiVO_4 \ (e^-) \tag{3}$$

$$BiVO_4 \ (h^+) + Ag_2CO_3 \rightarrow Ag_2CO_3 \ (h^+) + BiVO_4 \tag{4}$$

In addition, the photogenerated electrons on Ag surface will migrate to the conduction band of Ag$_2$CO$_3$, and these photogenerated electrons will further transfer from Ag$_2$CO$_3$ to the conduction band of BiVO$_4$. As shown in Equations (5) and (6):

$$Ag + h\upsilon \rightarrow Ag^* \tag{5}$$

$$Ag^* + Ag_2CO_3 \rightarrow Ag_2CO_3 \ (e^-) + Ag^{+*} \tag{6}$$

In this case, due to the conduction band of BiVO$_4$ is surrounded by many photogenerated electrons, which possesses strong reducibility. Therefore, the photogenerated electrons

on the conduction band of $BiVO_4$ can participate in the reduction of hexavalent chromium ions to trivalent chromium ions. As shown in Equations (7) and (8):

$$Ag_2CO_3 \ (h^+) + OH- /H_2O \rightarrow \bullet OH \tag{7}$$

$$OTH + \bullet OH/h^+ \rightarrow products \rightarrow CO_2 + H_2O \tag{8}$$

Besides, many photogenerated holes are enriched in the valence band of $Ag_2CO_3$ and are captured by $H_2O$ and $OH-$ on the surface of the photocatalyst to form $\bullet OH$. Therefore, they are directly involved in the degradation of OTH.

## 3. Materials and Discussion

### 3.1. Materials

$Bi(NO_3)_3 \cdot 5H_2O$ (purity 99%), $Na_3VO_4 \cdot 12H_2O$ (purity 99%), $AgNO_3$ (purity 99%), $NaHCO_3$ (purity 99%) and all chemicals were analytically pure (purity 99%). The chemical reagents used were purchased from Sinopharm Chemical Reagent Co., Ltd. (Shanghai, China).

### 3.2. Preparation of Flower Globular BiVO₄

We studied the synthesized sample with reference to previous experiments [55]. First, $Bi (NO_3)_3 \cdot 5H_2O$ (60 mg) was added to ultrapure water (40 mL), followed by 5 min ultrasound treatment. Then $Na_3VO_4 \cdot 12H_2O$ (100 mg) solution was added, and the mixture turns yellow. It was finally transferred to a Telfon-lined stainless steel autoclave and heated for 8 h at 160 °C. After cooling the solid powder to room temperature, the $BiVO_4$ sample was successfully prepared by washing with ultrapure water and ethanol several times.

### 3.3. Synthesized of Ag₂CO₃/BiVO₄ Heterojunction Photocatalyst

Firstly, 1 mmol $BiVO_4$ was dissolved in 50 mL ultrapure water and treated with ultrasound treatment for 5 min. In addition, $AgNO_3$ (0.20 mmol) solution was put in the mixture and stirred at a constant speed for about 30 min, then 0.10 mmol $NaHCO_3$ solution was added to make the pH neutral. Finally, the mixed solution was placed in the dark for 7 h and centrifuged to obtain the $Ag_2CO_3/BiVO_4$ samples.

### 3.4. Preparation of Ag/Ag₂CO₃/BiVO₄ Photocatalyst

First, $Ag_2CO_3/BiVO_4$ (0.5 g) was put in ethanol (50 mL), then 5 min ultrasonic treatment, and then exposure to ultraviolet light for 3 h. Finally, the final samples were obtained by centrifugation.

### 3.5. Photocatalytic Procedures

The catalytic performance of the samples for degradation of OTH was investigated using a 500 W Xe lamp (100 $mW/cm^2$) with a cut-off wavelength of $\leq$420 nm. The weight of catalyst was 0.4 g/L, the original concentration of OTH and $Cr^{6+}$ is 20 and 15 mg/L, respectively. Before illumination, the mixture was stirred in the dark for 60 min to reach an adsorption-desorption equilibrium. After exposure to visible light for different times, take some mixed solutions and measure the pollutant concentration after centrifuging to remove the powder samples.

### 3.6. Characterization

In this paper, X-ray diffraction (XRD), Field emission scanning electron microscope (FESEM), X-ray photoelectron spectra (XPS), photoluminescence spectroscopy (PL) and transmission electron microscope (TEM) etc. The specific details of these characterized instruments, SEM-EDS elemental mapping of $Ag/Ag_2CO_3/BiVO_4$ (Figure S1) and $N_2$-adsorption isotherms of $Ag/Ag_2CO_3/BiVO_4$ (Figure S2) will be supplemented in the supplementary materials.

## 4. Conclusions

A novel $Ag/Ag_2CO_3/BiVO_4$ plasmonic photocatalyst was achieved by depositing Ag nanoparticles on the surface of $Ag_2CO_3/BiVO_4$ through the photoreduction reaction. The morphology, structure and optical properties of the prepared samples were characterized by XRD and XPS, FESEM and TEM, UV–vis and PL. The catalytic properties of the sample were investigated by removing $Cr^{6+}$ and degradation of OTH, and the degradation rate of OTH reached 98.0% after 150 min of illumination. Comparing $BiVO_4$, $Ag_2CO_3/BiVO_4$ and $Ag/Ag_2CO_3/BiVO_4$, the latter possess excellent photocatalytic performance by enhancing light utilization and inhibiting the recombination by charge transfer and electron-hole pairs in $Ag/Ag_2CO_3/BiVO_4$. In addition, the results showed that $h^+$ and $\bullet$OH were the main active species in the photocatalytic degradation of OTH, and the enhanced photocatalytic activity mechanism of $Ag/Ag_2CO_3/BiVO_4$ photocatalyst was systematically studied in terms of degradation of OTH and reduction of $Cr^{6+}$.

**Supplementary Materials:** The following supporting information can be downloaded at: https://www.mdpi.com/article/10.3390/catal12121498/s1, Characterization; Analytical methods; Figure S1: SEM-EDS elemental mapping of $Ag/Ag_2CO_3/BiVO_4$; Figure S2: $N_2$-adsorption isotherms of $Ag/Ag_2CO_3/BiVO_4$.

**Author Contributions:** J.C. designed this work and prepared the manuscript. M.Z., X.Y., X.Z., Y.Y. and Y.L. carried out the experiments and data analysis. H.Z. and W.Z. participated in the discussion of the results. All authors have read and agreed to the published version of the manuscript.

**Funding:** This research received no external funding.

**Data Availability Statement:** The study did not report any data.

**Acknowledgments:** The authors greatly acknowledge the National Natural Science Foundation of China (No. 41573061, 51578279 and 51278242), the China Postdoctoral Science Foundation funded the 61th batches surface (2017M610336), and the Shanghai Tongji Gao Tingyao Environmental Science & Technology Development Foundation (STGEF).

**Conflicts of Interest:** The authors declare no conflict of interest.

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
