# Peer review of "Fabrication of a Plasmonic Heterojunction for Degradation of Oxytetracycline Hydrochloride and Removal of Cr(VI) from Water"

_catalysts, doi:10.3390/catal12121498_

Round 1
Reviewer 1 Report
The manuscript by Zhao W. and co-workers describes the synthesis and characterization of a novel plasmonic photocatalyst and its behaviour in the degradation of OTH and removal of Cr6+. The topic is very interesting, of great importance for the development of novel plasmonic heterojunction for different applications.
However, I have some concern in relation to certain aspects of the work, that which should be resolved by the authors before publication in Catalysts
-The supplementary material does not correspond to the work presented, for example BET, FESEM, they are techniques that have not been commented on in the main text. The samples have been deposited on ITO electrodes?? It's pretty confusing, and doesn't help.
The authors must indicate the meaning of OTH, AO, IPA … in the text, for better understanding.
In figure 1, the authors must indicate in the illustration, the composition of each step in the formation of the photocatalyst.
The authors must include in the experimental part:
1. What technique have they used to calculate the concentrations?
2. The calculation used to obtain the band gap values (can be included in the SI if the authors prefer)
3. a section indicating how the different recycling of the photocatalyst has been carried out.
In figure 6 c and 6f, shouldn't the x-axis correspond to runs instead of time? Could the authors explain in more depth how the different experiments have been carried out?
Author Response
- Answers to the first reviewer’s questions (Reviewer #3)
Question 1: The supplementary material does not correspond to the work presented, for example BET, FESEM, they are techniques that have not been commented on in the main text. The samples have been deposited on ITO electrodes It's pretty confusing, and doesn't help.
Answer: Thank you very much for your prompt attentions and intensive criticisms to our manuscript. We are sorry for the confusion caused to you. For BET, we have supplemented the data in the supplementary materials. SEM in this paper should be FESEM, we have simplified it. According to the reviewer's opinion, SEM in the paper has been modified to FESEM. As for the ITO electrode in the photoelectrochemical characterization, in fact, the ITO electrode was prepared to measure the photocurrent.
BET is shown in Fig. S2:
Fig. S2 N2-adsorption isotherms of Ag/Ag2CO3/BiVO4
According to the measurement results, the specific surface area of Ag/Ag2CO3/BiVO4 is large, and its value is 55.6 m2/g.
Question 2: The authors must indicate the meaning of OTH, AO, IPA … in the text, for better understanding.
Answer: We sincerely appreciate your detailed and professional advices for the improvement of the current paper. In order to make readers better understand the meaning of OTH, AO and IPA, we have explained in the paper and marked it in red.
oxytetracycline hydrochloride (OTH)
ammonium oxalate (AO)
isopropanol (IPA)
the local surface plasmon resonance (LSPR)
4-hydroxy-2,2,6,6-tetramethylpiperidine-N-oxyl (TEMPOL)
Question 3: In figure 1, the authors must indicate in the illustration, the composition of each step in the formation of the photocatalyst.
Answer: Thanks a lot for you to raise the question. We have recreated Fig.1 to show the composition of each step in the photocatalyst formation process.
Fig. 1 Schematic illustration of the synthesis process of Ag/Ag2CO3/BiVO4 photocatalyst.
Question 4: The authors must include in the experimental part:
- What technique have they used to calculate the concentrations?
Answer: Thank you very much for your insightful suggestions. The concentration of OTH in solution was determined by UV–vis spectroscopy. We will specify this in the supplementary materials.
The sentence “The concentration of OTH in solution was determined by UV–vis spectroscopy. The concentration of Cr6+ in solution was also determined by UV–vis spectroscopy using diphenylcarbazide reagent as a developer. The photocatalytic oxidation efficiency of OTH and the reduction efficiency of Cr6+ were calculated from the following expression:
η = (Ct - C0)/ C0 × 100%
where η is the photocatalytic degradation efficiency; Ct and C0 are the concentration of OTH (or Cr6+) before and after photocatalytic reaction.” has been added in the revised manuscript.
- The calculation used to obtain the band gap values (can be included in the SI if the authors prefer)
Answer: It is greatly appreciated of you to raise this question. We calculate the band gap value according to the following references.
H. Xu, H. Li, C. Wu, J. Chu, Y. Yan, H. Shu, Z. Gu, Preparation, characterization and photocatalytic properties of Cu-loaded BiVO4, J. Hazard. Mater. 153 (2008) 877-884.
- a section indicating how the different recycling of the photocatalyst has been carried out. In figure 6c and 6f, shouldn't the x-axis correspond to runs instead of time? Could the authors explain in more depth how the different experiments have been carried out?
Answer: Thanks a lot for your careful and detailed advice. In Figure 6c and 6f, the x-axis corresponds to run, which represents the number of cycles. Details of the cycle test are as follows:After the photocatalytic reaction, the suspension was centrifuged to obtain the powder photocatalysts, then, the powder photocatalysts were washed with deionized water and the dried sample was used for the next cycle test.

Reviewer 2 Report
This manuscript exhibits the fabrication Ag/Ag2CO3/BiVO4 plasmonic photocatalyst. The catalytic performance was examined on the degradation of oxytetracycline hydrochloride and removal of Cr (VI) ion. The synthesized photocatalyst was characterized. Interesting results were presented, and the topic is within the scope of journal of catalysts. However, the manuscript should be revised according to the following suggestion. My comments regarding the main points of the paper are as follows:
1. The title should be revised as: Fabrication of a plasmonic heterojunction for degradation of oxytetracycline hydrochloride and removal of Cr (VI) from water.
2. The full name of abbreviations should be given when they first appeared in the paper. Please revise abstract.
3. Keywords: additional keywords are recommended
4. Introduction: for first paragraph 9 citation reference is too much. The same for line 33 in the introduction. The authors should mention heavy metal and pharmaceutical residue water contamination in the introduction. key references:
https://doi.org/10.3390/coatings9080465, https://doi.org/10.1016/j.cej.2016.10.137 and doi:10.1088/1755-1315/612/1/012023.
5. Materials: please provide the purity of materials and reagent.
6. The authors should give the value of the irradiance (incident light intensity of the used lamp in mW/cm2)
7. The quality of some figures is low. Please improve it. All figure captions need to be improved. Please provide a full description of the presented data of the degradation and error bars should be added.
8. Please provide the analysis method for measuring tested pollutants' initial and residual concentrations.
9. The Discussion is too superficial and not deep enough. More close connections and quantitative results should be established and discussed.
10. Authors are recommended to revise the whole manuscript for language proof. Please pay attention to revising the language.
Author Response
- Answers to the second reviewer’s questions (Reviewer #2)
Question 1: The title should be revised as: Fabrication of a plasmonic heterojunction for degradation of oxytetracycline hydrochloride and removal of Cr (VI) from water.
Answer: Thanks a lot for your careful and detailed advice. We have revised the original title to: “Fabrication of a plasmonic heterojunction for degradation of oxytetracycline hydrochloride and removal of Cr (VI) from water”.
Question 2: The full name of abbreviations should be given when they first appeared in the paper. Please revise abstract.
Answer: Thanks a lot for you to raise the question. As for the full name of the abbreviations, we have modified it in the abstract and marked it in red. As follow:
oxytetracycline hydrochloride (OTH)
Question 3: Keywords: additional keywords are recommended
Answer: Thanks a lot for your careful and detailed advice. As for keywords, we are very willing to use other keywords and make substitutions in the paper. As follow:
Keywords: Photocatalysis; Photocatalyst; Photocatalytic performance; Heterojunction; Degradation.
Question 4: Introduction: for first paragraph 9 citation reference is too much. The same for line 33 in the introduction. The authors should mention heavy metal and pharmaceutical residue water contamination in the introduction. key references:
https://doi.org/10.3390/coatings9080465,https://doi.org/10.1016/j.cej.2016.10.137 and doi:10.1088/1755-1315/612/1/012023.
Answer: It is greatly appreciated of you to raise this question. We have cited the following literature in the paper.
[6] K.H.H. Aziz, K.M. Omer, A. Mahyar, H. Miessner, S. Mueller, D. Moeller, Application of Photocatalytic Falling Film Reactor to Elucidate the Degradation Pathways of Pharmaceutical Diclofenac and Ibuprofen in Aqueous Solutions, Coatings, 9 (2019) 465.
[7] K.H.H. Aziz, H. Miessner, S. Mueller, D. Kalass, D. Moeller, I. Khorshid, M.A.M. Rashid, Degradation of pharmaceutical diclofenac and ibuprofen in aqueous solution, a direct comparison of ozonation, photocatalysis, and non-thermal plasma, Chem. Eng. J. 313 (2017) 1033-1041.
[8] S.M. Abdulla, D.M. Jamil, K.H.H. Aziz, Investigation in heavy metal contents of drinking water and fish from Darbandikhan and Dokan Lakes in Sulaimaniyah Province - Iraqi Kurdistan Region, 612 (2020).
Question 5: Materials: please provide the purity of materials and reagent
Answer: It is greatly appreciated of you to raise this question. As follows:
Bi(NO3)3·5H2O (purity 99%), Na3VO4·12H2O (purity 99%), AgNO3 (purity 99%), NaHCO3 (purity 99%) and all chemicals were analytically pure (purity 99%). The chemical reagents used were purchased from Sinopharm Chemical Reagent Co.,Ltd.
Question 6: The authors should give the value of the irradiance (incident light intensity of the used lamp in mW/cm2)
Answer: Thanks a lot for you to raise the question. The irradiance value of xenon lamp we use is 100 mW/cm2.
Question 7: The quality of some figures is low. Please improve it. All figure captions need to be improved. Please provide a full description of the presented data of the degradation and error bars should be added.
Answer: It is greatly appreciated of you to raise this question. We have revised the fuzzy graph, and added error bars to the complete description of the existing data of degradation.
Fig. 6 (a) Photocatalytic activities of the samples and (b) pseudo-first-order kinetics of OTH degradations, (c) five recycling runs of OTH degradations, (d) the effect of different quenchers on the degradation of OTH
Fig. 7 (a) photocatalytic reduction activities of Cr6+ reduction, (b) five recycling runs of Cr6+ reduction.
Fig. 8 TOC removal of OTH for the different samples.
Question 8: Please provide the analysis method for measuring tested pollutants' initial and residual concentrations.
Answer: Thank you very much for your insightful suggestions. The concentration of OTH in solution was determined by UV–vis spectroscopy. We will specify this in the supplementary materials.
The sentence “The concentration of OTH in solution was determined by UV–vis spectroscopy. The concentration of Cr6+ in solution was also determined by UV–vis spectroscopy using diphenylcarbazide reagent as a developer. The photocatalytic oxidation efficiency of OTH and the reduction efficiency of Cr6+ were calculated from the following expression:
η = (Ct - C0)/ C0 × 100%
where η is the photocatalytic degradation efficiency; Ct and C0 are the concentration of OTH (or Cr6+) before and after photocatalytic reaction.” has been added in the revised manuscript.
Question 9: The Discussion is too superficial and not deep enough. More close connections and quantitative results should be established and discussed.
Answer: Thanks a lot for your careful and detailed advice. We will further discuss the discussion and results in this paper.
As shown in Fig. 9, the photocatalytic degradation mechanism of Ag/Ag2CO3/BiVO4 photocatalyst was proposed. Under visible light irradiation, photogenerated electrons and photogenerated holes are generated on the conduction band and valence band of Ag2CO3 and BiVO4, respectively. As shown in equation (1,2):
BiVO4 + hυ → BiVO4 (e- + h+) (1)
Ag2CO3 + hυ → Ag2CO3 (e- + h+) (2)
Since the conduction band (CB) and valence band (VB) position of BiVO4 are lower than Ag2CO3, the conduction band of photogenerated electrons transferred from Ag2CO3 to BiVO4, nevertheless the valence band of photogenerated holes are transferred from BiVO4 to Ag2CO3. As shown in equation (3,4):
Ag2CO3 (e-) + BiVO4 → Ag2CO3 + BiVO4 (e-) (5)
BiVO4 (h+) + Ag2CO3 → Ag2CO3 (h+) + BiVO4 (6)
In addition, the photogenerated electrons on Ag surface will migrate to the conduction band of Ag2CO3, and these photogenerated electrons will further transfer from Ag2CO3 to the conduction band of BiVO4. As shown in equation (5,6):
Ag + hυ → Ag* (3)
Ag* + Ag2CO3 → Ag2CO3 (e-) + Ag+* (4)
In this case, due to the conduction band of BiVO4 is surrounded by many photogenerated electrons, which possesses strong reducibility. Therefore, the photogenerated electrons on the conduction band of BiVO4 can participate in the reduction of hexavalent chromium ions to trivalent chromium ions. As shown in equation (7,8):
Ag2CO3 (h+) + OH-/H2O → •OH (7)
OTH + •OH/h+ → products → CO2 + H2O (8)
Besides, many photogenerated holes are enriched in the valence band of Ag2CO3 and are captured by H2O and OH− on the surface of the photocatalyst to form •OH. Therefore, they are directly involved in the degradation of OTH.
Question 10: Authors are recommended to revise the whole manuscript for language proof. Please pay attention to revising the language.
Answer: Thank you very much for your helpful advices. According to your suggestion, we have revised the entire manuscript in the appropriate language, and have marked red in the text.
- Answers to the second reviewer’s questions (Reviewer #2)
Question 1: The title should be revised as: Fabrication of a plasmonic heterojunction for degradation of oxytetracycline hydrochloride and removal of Cr (VI) from water.
Answer: Thanks a lot for your careful and detailed advice. We have revised the original title to: “Fabrication of a plasmonic heterojunction for degradation of oxytetracycline hydrochloride and removal of Cr (VI) from water”.
Question 2: The full name of abbreviations should be given when they first appeared in the paper. Please revise abstract.
Answer: Thanks a lot for you to raise the question. As for the full name of the abbreviations, we have modified it in the abstract and marked it in red. As follow:
oxytetracycline hydrochloride (OTH)
Question 3: Keywords: additional keywords are recommended
Answer: Thanks a lot for your careful and detailed advice. As for keywords, we are very willing to use other keywords and make substitutions in the paper. As follow:
Keywords: Photocatalysis; Photocatalyst; Photocatalytic performance; Heterojunction; Degradation.
Question 4: Introduction: for first paragraph 9 citation reference is too much. The same for line 33 in the introduction. The authors should mention heavy metal and pharmaceutical residue water contamination in the introduction. key references:
https://doi.org/10.3390/coatings9080465,https://doi.org/10.1016/j.cej.2016.10.137 and doi:10.1088/1755-1315/612/1/012023.
Answer: It is greatly appreciated of you to raise this question. We have cited the following literature in the paper.
[6] K.H.H. Aziz, K.M. Omer, A. Mahyar, H. Miessner, S. Mueller, D. Moeller, Application of Photocatalytic Falling Film Reactor to Elucidate the Degradation Pathways of Pharmaceutical Diclofenac and Ibuprofen in Aqueous Solutions, Coatings, 9 (2019) 465.
[7] K.H.H. Aziz, H. Miessner, S. Mueller, D. Kalass, D. Moeller, I. Khorshid, M.A.M. Rashid, Degradation of pharmaceutical diclofenac and ibuprofen in aqueous solution, a direct comparison of ozonation, photocatalysis, and non-thermal plasma, Chem. Eng. J. 313 (2017) 1033-1041.
[8] S.M. Abdulla, D.M. Jamil, K.H.H. Aziz, Investigation in heavy metal contents of drinking water and fish from Darbandikhan and Dokan Lakes in Sulaimaniyah Province - Iraqi Kurdistan Region, 612 (2020).
Question 5: Materials: please provide the purity of materials and reagent
Answer: It is greatly appreciated of you to raise this question. As follows:
Bi(NO3)3·5H2O (purity 99%), Na3VO4·12H2O (purity 99%), AgNO3 (purity 99%), NaHCO3 (purity 99%) and all chemicals were analytically pure (purity 99%). The chemical reagents used were purchased from Sinopharm Chemical Reagent Co.,Ltd.
Question 6: The authors should give the value of the irradiance (incident light intensity of the used lamp in mW/cm2)
Answer: Thanks a lot for you to raise the question. The irradiance value of xenon lamp we use is 100 mW/cm2.
Question 7: The quality of some figures is low. Please improve it. All figure captions need to be improved. Please provide a full description of the presented data of the degradation and error bars should be added.
Answer: It is greatly appreciated of you to raise this question. We have revised the fuzzy graph, and added error bars to the complete description of the existing data of degradation.
Fig. 6 (a) Photocatalytic activities of the samples and (b) pseudo-first-order kinetics of OTH degradations, (c) five recycling runs of OTH degradations, (d) the effect of different quenchers on the degradation of OTH
Fig. 7 (a) photocatalytic reduction activities of Cr6+ reduction, (b) five recycling runs of Cr6+ reduction.
Fig. 8 TOC removal of OTH for the different samples.
Question 8: Please provide the analysis method for measuring tested pollutants' initial and residual concentrations.
Answer: Thank you very much for your insightful suggestions. The concentration of OTH in solution was determined by UV–vis spectroscopy. We will specify this in the supplementary materials.
The sentence “The concentration of OTH in solution was determined by UV–vis spectroscopy. The concentration of Cr6+ in solution was also determined by UV–vis spectroscopy using diphenylcarbazide reagent as a developer. The photocatalytic oxidation efficiency of OTH and the reduction efficiency of Cr6+ were calculated from the following expression:
η = (Ct - C0)/ C0 × 100%
where η is the photocatalytic degradation efficiency; Ct and C0 are the concentration of OTH (or Cr6+) before and after photocatalytic reaction.” has been added in the revised manuscript.
Question 9: The Discussion is too superficial and not deep enough. More close connections and quantitative results should be established and discussed.
Answer: Thanks a lot for your careful and detailed advice. We will further discuss the discussion and results in this paper.
As shown in Fig. 9, the photocatalytic degradation mechanism of Ag/Ag2CO3/BiVO4 photocatalyst was proposed. Under visible light irradiation, photogenerated electrons and photogenerated holes are generated on the conduction band and valence band of Ag2CO3 and BiVO4, respectively. As shown in equation (1,2):
BiVO4 + hυ → BiVO4 (e- + h+) (1)
Ag2CO3 + hυ → Ag2CO3 (e- + h+) (2)
Since the conduction band (CB) and valence band (VB) position of BiVO4 are lower than Ag2CO3, the conduction band of photogenerated electrons transferred from Ag2CO3 to BiVO4, nevertheless the valence band of photogenerated holes are transferred from BiVO4 to Ag2CO3. As shown in equation (3,4):
Ag2CO3 (e-) + BiVO4 → Ag2CO3 + BiVO4 (e-) (5)
BiVO4 (h+) + Ag2CO3 → Ag2CO3 (h+) + BiVO4 (6)
In addition, the photogenerated electrons on Ag surface will migrate to the conduction band of Ag2CO3, and these photogenerated electrons will further transfer from Ag2CO3 to the conduction band of BiVO4. As shown in equation (5,6):
Ag + hυ → Ag* (3)
Ag* + Ag2CO3 → Ag2CO3 (e-) + Ag+* (4)
In this case, due to the conduction band of BiVO4 is surrounded by many photogenerated electrons, which possesses strong reducibility. Therefore, the photogenerated electrons on the conduction band of BiVO4 can participate in the reduction of hexavalent chromium ions to trivalent chromium ions. As shown in equation (7,8):
Ag2CO3 (h+) + OH-/H2O → •OH (7)
OTH + •OH/h+ → products → CO2 + H2O (8)
Besides, many photogenerated holes are enriched in the valence band of Ag2CO3 and are captured by H2O and OH− on the surface of the photocatalyst to form •OH. Therefore, they are directly involved in the degradation of OTH.
Question 10: Authors are recommended to revise the whole manuscript for language proof. Please pay attention to revising the language.
Answer: Thank you very much for your helpful advices. According to your suggestion, we have revised the entire manuscript in the appropriate language, and have marked red in the text.

Reviewer 3 Report
I went ahead and read the Jihui Cao et al. article with the title `` Fabrication of a plasmonic heterojunction for degradation of oxytetracycline hydrochloride and removal of hexavalent chromium ion" In terms of the findings, the paper presents intriguing possibilities. Overall, based on the research and the data, it is appropriate for publication. However, there are some issues that has to be fixed before publication.
- Need to mention the novelty of this work.
- Abstract also need to revise with results oriented.
· Need to mention the reference for synthesis process.
· Some errors regarding the sub/super script, spacing and typo need to consider throughout the manuscript.
· In introduction, mention some other related studies of other materials should be mentioned such as Energy and Environment Focus 2 (1), 73-78, Desalination and Water Treatment 46 (1-3), 205-214.
- The reference style should be consistent. Please check.
- Need to add EDX data of the material.
- Need to add BET data of the material to check the effect of the size.
- Conclusion also revised based on the results.
Author Response
- Answers to the third reviewer’s questions (Reviewer #1)
Question 1: Need to mention the novelty of this work. Abstract also need to revise with results oriented.
Answer: Thanks a lot for you to raise the question. The present study will benefit the development of the new plasmonic heterojunction photocatalysts and would be of great importance to meet the environmental demands in the future. Finally, the abstract is revised based on the results.
Question 2: Need to mention the reference for synthesis process.
Answer: Thank you very much for your insightful suggestions. We will add the references of the synthesis process to the paper, and the references are as follows:
[45] W. Zhao, B. Dai, F. Zhu, X. Tu, J. Xu, L. Zhang, S. Li, Y.C.L. Dennis, C. Sun, A novel 3D plasmonic p-n heterojunction photocatalyst: Ag nanoparticles on flflower-like p-Ag2S/n-BiVO4 and its excellent photocatalytic reduction and oxidation activities, Appli. Catal. Environ. 229 (2018) 171-180.
Question 3: Some errors regarding the sub/super script, spacing and typo need to consider throughout the manuscript.
Answer: Thanks a lot for your careful and detailed advice. We have carefully revised the sub/superscript, spacing and sorting errors of the whole article.
Question 4: In introduction, mention some other related studies of other materials should be mentioned such as Energy and Environment Focus 2 (1), 73-78, Desalination and Water Treatment 46 (1-3), 205-214.
Answer: It is greatly appreciated of you to raise this question. We will quote relevant studies of other materials as follows:
[10] M.M. Haque, A. Khan, K. Umar, N. A. Mir, M. Muneer, T. Harada, M. Matsumura, Synthesis, Characterization and Photocatalytic Activity of Visible Light Induced Ni-Doped TiO2, Energy Envir. Focus 2 (2013) 73-78.
[15] K. Umar, A.A. Dar, M.M. Haque, N.A. Mir, M. Muneer, Photocatalysed decolourization of two textile dye derivatives, Martius Yellow and Acid Blue 129, in UV-irradiated aqueous suspensions of Titania, Desalin. Water Treat. 46 (2012) 205-214.
Question 5: The reference style should be consistent. Please check.
Answer: Thank you very much for your kindly and helpful suggestions. We have revised the format of the references.
Question 6: Need to add EDX data of the material.
Answer: It is greatly appreciated that the reviewer raised this question. We have supplemented the EDS-mapping data of the materials in the supplementary materials.
Fig. S1 SEM-EDS elemental mapping of Ag/Ag2CO3/BiVO4.
The SEM-EDS element mapping of Ag/Ag2CO3/BiVO4 in Fig. S1 shows that it is composed of Bi, O, C, Ag and V elements.
Question 7: Need to add BET data of the material to check the effect of the size.
Answer: Thank you very much for your helpful advices. We have supplemented the BET data of the materials in the supplementary materials.
Fig. S2 N2-adsorption isotherms of Ag/Ag2CO3/BiVO4
According to the measurement results, the specific surface area of Ag/Ag2CO3/BiVO4 is the larger, and its value is 55.6 m2/g.
Question 8: Conclusion also revised based on the results.
Answer: Thanks a lot for your careful and detailed advice. We have revised the conclusion according to the experimental results. As follow:
- The morphology, structure and optical properties of the prepared samples were characterized by XRD and XPS, FESEM and TEM, UV and PL. The catalytic properties of the sample were investigated by removing Cr6+ and degradation of OTH, and the degradation rate of OTH reached 98.0% after 150 min of illumination.
- In addition, the results showed that h+ and •OH were the main active species in the photocatalytic degradation of OTH, and the enhanced photocatalytic activity mechanism of Ag/Ag2CO3/BiVO4 photocatalyst was systematically studied in terms of degradation of OTH and reduction of Cr6+.

Round 2
Reviewer 2 Report
The manuscript has been revised well and the quality of the revised manuscript has been improved. I suggest acceptance.